# Connecting the World of Healthcare Virtually: A Scoping Review on Virtual Care Delivery

**DOI:** 10.3390/healthcare9101325

**Published:** 2021-10-05

**Authors:** Cindy (Zhirui) Li, Elizabeth M. Borycki, Andre W. Kushniruk

**Affiliations:** 1School of Health Information Science, University of Victoria, Victoria, BC V8P 5C2, Canada; 2Michael Smith Foundation for Health Research, Vancouver, BC V6H 3X8, Canada

**Keywords:** virtual care, virtual clinics, healthcare, virtual reality

## Abstract

Virtual care extends beyond the walls of healthcare organizations to provide care at a distance. Although virtual care cannot be regarded as a solution for all health-related inquiries, it provides another care delivery channel for specific patient populations with appointments that do not require in-person physical examinations or procedures. A scoping review was conducted to define the meaning of virtual care, understand how virtual care has influenced the healthcare industry and is being expanded to complement the existing healthcare system, and describe the outcomes of using virtual care for patients and providers. Findings from the scoping review suggest that virtual care encompasses the provision of care using advanced video conferencing technology to support remote care that takes place between patients and providers and the use of virtual reality technology to simulate care environments. Some of virtual care’s use in healthcare includes application to pain and anxiety management, virtual consultations and follow-up visits, rehabilitation and therapy services, outpatient clinics, and emergency services. Lastly, from a provider and patient perspective, while both saw benefits of virtual care and scored the service relatively high on satisfaction after using virtual care, the greatest barrier to using virtual care may be technological challenges.

## 1. Introduction

Self-management of health is an essential part of everyday life. Historically, provider-centric models have been used by healthcare organizations (e.g., hospitals). In provider-centric models, patients seek care by visiting a provider’s office or making trips to healthcare facilities. For example, a patient wishing to receive diabetes education for insulin management would need to visit a diabetes education clinic with data from their glucometers to show a registered nurse (RN), dietician, or physician so that they are able to receive the necessary support from their providers. With technological advances in healthcare, there is an opportunity to enhance existing patient–provider relationships by creating new tools to support a patient-centric model of care. This patient-centric model of care occurs when providers and healthcare organizations are trying to provider better access to care from a patient’s perspective, which can include offering services such as telehealth visits. Virtual care provides the opportunity for the description above to become a reality, as virtual care is able to increase the level of convenience for patients to access care from their providers.

Many technologies exist in the global communications market to support virtual connections through real-time asynchronous communication, such as Skype^®^ or FaceTime^®^. These technologies connect family and friends regardless of geographic location and/or time differences. When it comes to healthcare, there are privacy concerns in using these common tools in terms of Personal Identifiable Information (PII) and Protected Health Information (PHI). PII refers to information that recognizes individuals based on unique identifiers, such as a name, social insurance number, and/or driver’s license [1]. A person’s PHI refers to information generated from medical records, such as name, medical record numbers, or biometric identifiers, but it may also contain many of the identifiers in PII, such as name, driver’s license number, or passport number [1]. With this in mind, companies from all around the globe have developed tools specifically for use in healthcare. These tools allow for compliance with local privacy regulations; for example, companies in the U.S. comply with the Health Insurance Portability and Accountability Act (HIPAA) [2]. In Canada, virtual tools comply with the Freedom of Information and Protection of Privacy Act (FIPPA) according to the Office of the Information and Privacy Commissioner and the Personal Information Protection and Electronic Documents Act (PIPEDA) [3]. Hence, around the globe, countries have different regulations that vendors must comply with in order to produce technologies that are acceptable by the privacy and security standards of these various countries. These acts serve the purpose of protecting an individual’s private information from wrongful use or distribution into the black hole of our information-heavy world.

As the use of many of these tools does not require the physical presence of both the patient and provider, virtual care has become the generic term used to refer to this model of care. Although virtual care is not a new concept, it is not well understood by the general public in terms of the types of the services virtual care can offer or how virtual care services are provided compared to traditional telehealth or even face-to-face visits. The major difference between these three modalities of services is the location of patients. Face-to-face visits are obvious in that the patient would be collocated with their provider, whether it is in their provider’s office, or a provider in the community, such as a community health worker who visits a patient in the patient’s home. For traditional telehealth, it means that a patient will be attending their remote consultation with their provider from a healthcare facility through a designated space where video conferencing equipment has been set up on both the patient and provider’s side to have a video consultation. Finally, virtual care in this particular context means that the patients and providers can be anywhere, such as in the comfort of their own home rather than having to travel to the hospital. Hence, the objective of this research is to explore the world of virtual care and the tools that are specifically designed to be applied to healthcare services. This study will (1) define the meaning of virtual care; (2) understand how virtual care has influenced the healthcare industry; (3) understand how virtual care is being expanded to complement the existing healthcare system, and (4) describe the outcomes of using virtual care for patients and providers.

## 2. Materials and Methods

The protocol followed for this scoping review aligns with a five-step framework outlined by Arksey and O’Malley, and advanced by Levac et al. [4]. The framework includes identifying the research questions and relevant studies, selecting studies, data extraction from the selected articles, and a dissemination of the results [4]. An ethics review was completed at the University of Victoria by the Research Ethics Coordinator and Research Ethics Vice-Chair from the Office of Research Services. Upon review, it was deemed that this project does not involve human subjects and is limited to the use of publicly available data or articles; hence, it is exempt from a full human research ethics review.

### 2.1. Identifying Research Questions

The scoping review answered the following research questions:How is virtual care defined?How has virtual care influenced the healthcare industry?How is virtual care being expanded to complement the existing healthcare system?What are the outcomes of virtual care for patients’ and caregivers’ perspectives?

### 2.2. Identifying Relevant Studies

As part of this scoping review, literature from the following databases were searched: Medline^®^, PubMed^®^, CINAHL^®^, IEEE Xplore^®^, ISI Web of Science^®^, Biomed Central^®^, Applied Science^®^, and Technology Index^®^. Two important search criteria to note are publication dates and keywords. The published literature from the past 10 years was reviewed in depth and breadth in the topic area (i.e., virtual care). Keywords that were searched in these databases include but are not limited to those shown in the Table 1 below.

### 2.3. Selecting Studies

Covidence^®^ is the online platform used for systematic review management. Zotero^®^ is an electronic reference management system used to store all potential studies to be included in this scoping review. Zotero^®^ provides the ability to store all studies in a virtual library, which can then be imported into Covidence^®^ to complete an initial title and abstract screening. Following a title and abstract screening in Covidence^®^, a number of irrelevant articles were eliminated, resulting in a list of articles that would require full-text reviews. Two investigators were involved with approving articles for inclusion in the scoping review. A meeting was set up via video conference to discuss any discrepancies between the investigators when deciding to include or exclude an article. Articles were included if they touched on at least one of the research questions and involved the use of virtual care technology at a distance, and excluded otherwise. Articles were also excluded if they were not a full study. As a result, a PRISMA diagram was produced, visually depicting the process of elimination. Duplicate articles were automatically removed from Covidence^®^ if they had the same author and title. Studies deemed irrelevant were summarized with the reasons for their exclusion. Following the selection of studies, data charting was performed to list the authors, date, title, method, results, limitations, and any other fields as necessary for each article that was screened.

### 2.4. Data Charting

Data charting was completed on all 34 studies included from the PRISMA Diagram in Figure 1. Information extracted from each article included the: authors’ names, the publication date, title, method, results, limitations, and any other fields as necessary for each article that was screened. Full results of data charting are consolidated into tables below in Appendix A and Appendix B.

### 2.5. Dissemination of Results

As part of the data-charting process, themes for each article were documented in the table that relate to one or more of the four research questions this scoping review answers. These themes include consumer perspectives on virtual care, whether it be a provider or a patient, how virtual care can be defined, as well as applications of virtual care. Consumer perspectives include user experience, both from the provider side of care and from the receiver end. This helps inform the direction of virtual care in the future. Currently, virtual care is a relatively broad topic, as can be seen from the keywords table in Section 2.2; therefore, a selection of care services provided remotely, or are simulated, such as through virtual reality, can all fall under the umbrella of virtual care. Lastly, given the broad definition virtual care can encompass, applications of virtual care can also thrive or fail in various settings, such as pain management or follow-up visits. The results section below will further discuss the details uncovered from data charting with references.

## 3. Results

The PRISMA diagram generated from the elimination of articles throughout the screening process is shown in Figure 1 below.

**Figure 1 healthcare-09-01325-f001:**
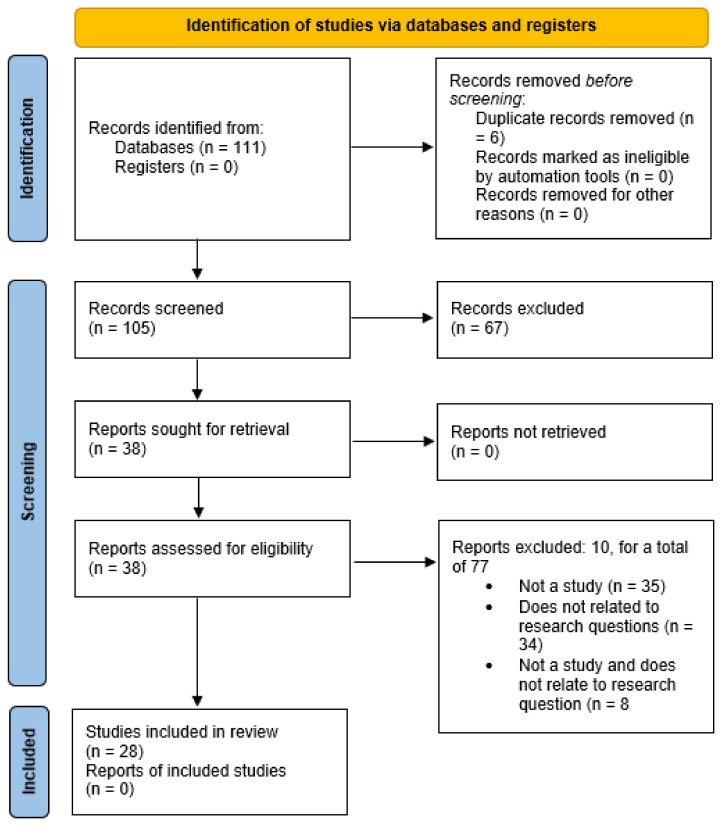
PRISMA diagram.

Given that the range of articles searched in this scoping review was relatively large, there were no search criteria that limited how recent an article had to be in order to be included in the project. This produced an interesting result in that the majority of articles included in this study were published within the last five years (≥015); only 29.41% (10/34) were published before 2015. Even including the percentages of excluded articles, that decreases to just 20.79% (21/101) of the articles on the topic of virtual care being published before 2015. Based on these statistics, it can be seen that more research on virtual care has been done in recent years. Then, the articles were reviewed with attention to the following themes: (1) the definition of virtual care, (2) the influence of virtual care on the healthcare industry, (3) the expansion of virtual care to complement the existing healthcare system, and (4) consumer perspectives of virtual care.

### 3.1. Definition of Virtual Care

Based on the selection of articles uncovered in this scoping review, virtual care can encompass the provision of care through using advanced video conferencing technology between patients and providers remotely or using virtual reality technology to simulate care environments. De Jong et al. [5] defined virtual care and its application as, “integrated web-based technology that combines self-management, data sharing, and communication between patients and professionals”. Others, such as Gordon et al. [6] also noted that virtual health can be defined as the utilization of real-time video consultation with a provider over the Internet, and that they expect the need for virtual care will continue to grow as the use of mobile devices and patient demand for immediate and convenient access to care increase. However, virtual reality is typically not defined as virtual care. Virtual reality is a tool that can support virtual care. This will be discussed in more detail in the following sections.

### 3.2. Influence of Virtual Care on the Healthcare Industry

In recent years, virtual care has expanded. It now touches on multiple aspects of healthcare, including pain and anxiety management, virtual consultations and follow-up visits, rehabilitation and therapy services, outpatient clinics, and emergency services.

#### 3.2.1. Pain and Anxiety Management

Virtual care has been applied to the management of pain and anxiety associated with the diagnosis of hematologic diseases; for example, the examination of bone marrow is an essential step in the diagnosis and management of hematologic diseases; hence, “patients undergoing bone marrow aspiration and biopsy procedures commonly experience pain, anxiety and stress” [7]. Nevertheless, distraction has proven to be an effective nonpharmacologic intervention to decrease pain and anxiety [7]. Virtual reality goggles were used by adults in an outpatient cancer center to determine whether the visual and auditory stimuli provided by the goggles “would decrease the amount of pain and anxiety experienced by patients during bone marrow aspiration and biopsy procedures, as compared to the practice of viewing a television with sound” [7]. Vital signs were collected from both groups; however, the results showed that there were no significant decreases in the level of pain experienced by the group wearing goggles versus the group watching television [7]. Although distraction through virtual reality did not prove effective in this scenario, other studies further investigate the appropriateness of using virtual reality in clinical environments.

According to Birnie et al. [8], “needle procedures are among the most distressing aspects of pediatric cancer-related treatment. Virtual Reality (VR) distraction offers promise for needle-related pain and distress given its highly immersive and interactive virtual environment.” The researcher studied the ease of use and acceptability of using a VR intervention with children diagnosed with cancer who were undergoing the implantation of an intravenous access device (IVAD) for needle insertion [8]. The researchers found that 82% of study participants found the intervention easy to use, 94% found the intervention easy to understand, and 94% indicated they would like to use the intervention for subsequent needle procedures [8]. In addition to this, the study findings revealed no adverse events such as nausea or dizziness were reported by the participants; hence, with the right refinements based on the need of each clinical procedure, VR can still provide an alternative intervention for pain management.

Pain intensity has also been observed with VR treatment in a study by Scapin [9]. Scapin [9] studied the effect of VR treatment upon the perception of pain in two burned children who were hospitalized in a Burn Treatment Center in Southern Brazil. “For assessing pain, a facial pain rating scale was applied at four times; just before the dressing, during the dressing without the use of VR, during the dressing with VR, and after the use of VR” [9]. The researcher found that the application of VR during burn treatments showed relevant effects, as the children were immersed in the VR environment during the dressing procedure [9]. This also meant that the amount of sedation that was originally required was also decreased; hence, VR could be used as a nonpharmacological intervention for pain management.

Schneider et al. [10] studied the use of VR as a distraction intervention to relieve the levels of distress in adults undergoing chemotherapy for breast, colon, and lung cancer. Similar to an earlier study, no significant decrease in stress levels was observed; however, 82% of respondents in their study preferred to use VR for future treatments. The study participants “stated that using VR made the treatment seem shorter and that chemotherapy treatments with VR were better than treatments without the distraction intervention” [10].

In summary VR has been used to distract individuals while they are undergoing painful treatments such as burn care. VR has also been successfully used to help individuals manage their anxiety. The approach holds considerable promise for use in managing pain and anxiety associated with other types of treatments. With the right amount of refinement to this technology, it can adapt to any clinical setting. Although this intervention may not result in significant levels of pain or anxiety reduction, it does help in making complex procedures less difficult to experience.

#### 3.2.2. Virtual Consultations and Follow-Up Visits

In a study by Abbot et al. [11], virtual consults were found “to expedite diagnostic and therapeutic interventions for veterans with incidentally discovered pulmonary nodules”. A total of 157 virtual consults were completed, and a comparison between virtual visits versus in-person visits spanned a duration of 6 months [11]. Findings from this study indicate a decrease both in cost and the time required to complete each visit virtually. “For all virtual consults, the mean time to completion of consultation was 3.2 days. Subsequent in-person consultations during the first 6 months of virtual consults occurred within a mean time of 20.5 days. The average thoracic surgery outpatient facility visit was $228 per in-person consultation, whereas the virtual consult cost was $120 per episode, a 47.4% decrease [11]. In another study, Schneider and Hood [12] compared telephone-based virtual outpatient clinics with traditional in-person outpatient clinics as they were used to follow up on general surgery patients. “Of 107 subjects randomized to virtual follow-up, 98 (92%) were successfully contacted by telephone, of which 10 (10%) had postoperative issues and 3 of whom ultimately attended a conventional clinic for follow-up. Of 102 subjects randomized to the conventional outpatient clinic follow-up, 83 (81%) attended the appointment, of which 16 (19%) had postoperative concerns” [12]. From both these studies, feedback has been positive, and most participants also expressed preference for virtual follow-up visits in the future.

As virtual healthcare continues to expand, Gordon et al. [6] examined the costs of virtual visits over a period of three weeks compared to in-person visits in retail health clinics, urgent care centers, emergency departments, or primary care physician visits. They applied a cross-sectional, retrospective analysis of insurance company claims with a total of 4635 virtual visits and 55,310 non-virtual visits included. The costs for retail health clinics, urgent care centers, emergency departments, and primary care physician visits were estimated to be $36, $153, $1735, and $162 higher in non-virtual visits than virtual visits, respectively [6]. Trends observed from these studies indicate an overall patient and healthcare provider satisfaction with virtual visits. In particular, follow-up visits may be one of the most appropriate appointment types to be conducted virtually, given that physical assessments are often not required. In addition to this, there was also a substantial decrease in the costs associated with completing virtual versus in-person visits.

#### 3.2.3. Rehabilitation and Therapy Services

Tele-rehabilitation allows patients to access rehabilitation services remotely through video conferencing in their own homes. Virtual reality (VR) has been applied to patient balance training in the past, “which [has] been shown to reduce postural instability in patients with Parkinson’s Disease (PD)” [13]. In the multi-centered study by Gandolfi et al. [13], 38 PD patients were assigned to receive telerehabilitation treatments using the Nintendo Wii Fit system in their homes, while another group of 38 PD patients were assigned to in-clinic sensory integration balance training (SIBT). Analysis revealed that “static and dynamic postural control was improved in the PD patients who had received in-home VR-based balance training (TeleWii), while improvements in mobility and dynamic balance were greater, on average, in those who had received in-clinic SIBT... In addition, the total cost of rehabilitation using TeleWii was lower than that of SIBT”. The study did lack instrumental evaluations to assess balance performance, postural reactions, and changes in muscle strength; however, virtual reality is still considered a feasible alternative to in-clinic SIBT [13].

In a similar study, researchers examined the “effects of VR augmented balance training on the sensory integration of postural control under varying attentional demands and compared the results to those in a conventional balance (CB) training group and an untrained control group.” A longitudinal, randomized controlled trial was used with sensory organization tests (SOTs), and verbal reactions times (VRTs) were recorded [14]. “Both VR and CB training improved sensory integration for postural control in people with PD, especially when they were deprived of sensory redundancy” [14].

Finally, a study by Klein et al. [15] described the application of virtual reality over 5 weeks to “improve gait and mobility in people with a history of falls, poor mobility, or postural instability.” In their study, a retrospective data analysis was conducted on clinical records of patients who were recruited to training by walking on treadmills while dodging virtual obstacles. The results after 5 weeks indicated time to completion increases in three tests—the Up and Go Test (TUG), Two-Minute Walk Test (2MWT), and Four-Square Step Test (FSST)—by 10.3%, 9.5%, and 13%, respectively. Therefore, the combination of treadmill training and virtual obstacles appear to be both practical and effective for physical therapy treatments, especially if patients do not need to leave the comfort of home [15]. As can be seen from the examples in the studies described above, there is an opportunity to successfully integrate remote rehabilitation interventions, also known as tele-rehabilitation, into healthcare. For patients who require this service, reducing the need to travel to an in-person clinic also provides greater convenience and an opportunity to improve physical function (if they are already physically challenged).

#### 3.2.4. Outpatient Care

A virtual outpatient clinic was established as a cloud-based, multicomponent outpatient clinic and studied by De Jong et al. [5] to assess the feasibility and functionality of virtual outpatient clinics. “The [virtual outpatient clinic] consists of 6 digital tools that facilitate self-monitoring (blood pressure, weight and pain) and communication with peers and providers (chat and videoconferencing) connected to a cloud-based platform and the hospital patient portal to facilitate access to (self-collected) medical data” [5]. Feasibility, adherence, usage statistics, technical issues identified, and qualitative responses were assessed, indicating that most participants successfully used all options of the virtual outpatient clinic and expressed positive attitudes to the use of the various tools within the virtual outpatient clinic. “The adherence was 7/19 for weight scale, 11/19 for blood pressure monitor, and 14/20 and 17/20 for pain score and daily questions, respectively. The adherence for personal health record was 13/20 and 12/20 for the patient portal system” [15]. The participants in this virtual outpatient clinic were more technically savvy. As a result, the findings from this study may not be generalized to the greater population, as people who are comfortable with more technology are more likely to accept the use of various tools within a virtual outpatient clinic.

Healey et al. [16] conducted a randomized trial aimed at determining whether a virtual outpatient clinic is an acceptable alternative to a traditional outpatient clinic for a range of general surgical patients’ post-discharge. All patients who were admitted under one general surgical service over the span of the study period were assessed and randomized to receive either virtual outpatient clinic or outpatient clinic appointments. The following points summarize the results obtained from a follow-up questionnaire sent to the participants:98/107 (91.6%) patients in the virtual outpatient clinic group were successfully contacted.83/102 (81.4%) patients in the outpatient clinic group were successfully contacted.10 patients in the virtual outpatient clinic group reported ongoing issues.6 patients in the outpatient clinic group reported ongoing issues.78/82 (95%) of patients in the virtual outpatient clinic group were happy with their overall experience.34/61 (56%) of patients in the outpatient clinic group were happy with their overall experience.68/81 (83%) of patients in the virtual outpatient clinic group preferred a virtual outpatient clinic appointment in the future.41/61 (67%) of patients in the outpatient clinic group preferred a virtual outpatient clinic appointment in the future.

It is standard practice to follow-up with patients from surgical discharge; however, the demand and resource balance typically tips to the demand side. Therefore, a virtual outpatient clinic can help reduce unnecessary in-clinic visits to provide the resources to patients who are more in need of physical care.

#### 3.2.5. Emergency Services

Research suggests that VR may be useful in the context of emergency services such as those provided to children who have sickle cell anemia when they experience a vaso-occlusive crisis. In America, approximately 100,000 of the total population are affected by sickle cell disease, and about 300,000 infants are born with sickle cell disease each year. “Children require prescriptions for pain medication, often opioids, during a vaso-occlusive crisis experienced at home, but they seek medical care in the emergency department when the pain is not improving” [17]. In their study, the investigators attempted to determine the effects of using virtual reality on acute pain in pediatric patients with sickle cell disease experiencing a vaso-occlusive crisis. A sample of 15 participants were either assigned to a control group that received standard intravenous narcotics treatments or an intervention group that received virtual reality along with standard treatment [17]. “Pain was assessed using the Numerical Rating Scale (NRS) and the Face, Legs, Activity, Cry, Consolability (FLACC) scale. Data were analyzed using an independent sample *t* test” [17]. The results suggest that although there was no significant statistical difference between the pain scores of both control and intervention groups, the study does help confirm that introducing nonpharmacologic treatments may be beneficial for pediatric patients experiencing pain, as it is a distraction technique to help ease pain [17].

### 3.3. Expansion of Virtual Care to Complement the Existing Healthcare System

Virtual care, in all shapes and forms, have also been used in health professional training and education (in addition to patient care). VR has been used for remote training of healthcare professionals. Remote training has taken the form of online training platforms, virtual reality, as well as use of virtual patients.

#### 3.3.1. Online Training Platforms

In a study by Tong et al. [18], a post-treatment investigation of the effectiveness of five automated “self-help cognitive behavior e-therapy programs for generalized anxiety disorder (GAD), panic disorder with or without agoraphobia (PD/A), obsessive–compulsive disorder (OCD), posttraumatic stress disorder (PTSD), and social anxiety disorder (SAD) offered to the international public via Anxiety Online, an open-access full-service virtual psychology clinic for anxiety disorders” was undertaken. Participants (n = 255) were asked to evaluate each of five online therapy programs [18]. Overall, there were significant observed reductions on all five anxiety disorder clinical severity ratings, a decreased number of clinical diagnoses, and an increased confidence in the self-management of mental healthcare [18].

In a study conducted by Trong et al. [19], the researchers provided an overview of an online training program for health professionals aimed at enhancing their skills in supporting pregnant women attempting to stop smoking. The online program offers “5 interactive case simulations and comprehensive discussion of patient visits, short lectures on relevant topics from leading experts, interviews with real patients who have quit, and a dedicated website of pertinent links and office resources” [19]. The incentive for health professionals to undergo training is up to 4.5 h of continuing education credits, but other benefits include understanding the 5As (ask, advise, assess, assist and arrange) counseling approach [19]. “Although clinical guidelines consistently recommend screening and counseling by prenatal care providers, only half or pregnant smokers receive counseling... Reasons for the low level of cessation counseling include providers’ self-reported lack of awareness of or agreement with existing guidelines, lack of self-efficacy, lack of training, lack of systems to support counseling activities, and lack of patient and provider materials” [19]. Therefore, an online training program improves time management for both patients and providers (increases flexibility in scheduling training) and helps sharpen the skills of providers aimed at encouraging pregnant women to quit smoking and remain smoke-free post-partum.

#### 3.3.2. Virtual Reality

Virtual reality has been used for medical use cases, such as therapy treatments as discussed in earlier sections. “Applications of VR in pharmacy include adjunctive or replacement treatment for pharmacotherapy in pain management, anxiety and other disorders, pharmacological modeling for drug discovery, pharmacist education and training, and patient counselling and behavior modification” [20]. The results obtained from many clinical studies indicated that the majority of the subjects reported positive experiences with VR, as it alleviates stress and anxiety from certain illnesses through providing distraction [20]. Despite the positive results, Ventola [20] notes the fact that the younger populations tend to be more willing to use VR compared to older populations. However, this is understandable, as there can never be a solution that satisfies everyone.

Dyer’s research work [21] involved using the technology to teach medical students and other health professionals’ empathy toward older adults through virtual reality by allowing them to “simulate being a patient with age-related diseases and to familiarize medical students with information resources related to the health of older adults.” The software used in the study created an immersive virtual environment. Each virtual reality kit contained an Alienware^®^ laptop, Oculus Rift^®^ headset with a sensor and a Leap Motion^®^ hand-tracking device, which cost a total of $2000–$2500 [21]. As a result, not only did the project successfully introduce an innovative teaching modality, VR also “enhanced students’ understanding of age-related health problems and increased their empathy for older adults with vision and hearing loss or Alzheimer’s disease” [21]. This is a great segue into the next study, as [22] evaluate the use of virtual patients to supplement education for health professionals in pediatric dentistry.

#### 3.3.3. Virtual Patients

According to Papadopoulos et al. [22], a child virtual patient (VP) was programmed and called Erietta. Erietta is an 8-year-old girl who visits the dentist with her mother for the first time. “Communication techniques such as Tell–Show–Do and parents’ interference management were the basic elements of the education scenario on which the VP was based” [22]. A total of 103 dental students were placed into an experimental group, which was exposed to the simulation, and a control group, which did not receive the simulation. Both groups were asked in the end to complete a questionnaire following their participation in the research [22]. The majority of participants provided a positive evaluation of the simulation. There was a 69% satisfaction rate among the experimental group who expressed preference for using VP to supplement traditional training [22]. In another study by Papadopoulos et al. [23], the researchers created a virtual patient to support learning processes; however, this time, the virtual patient that was created was specific to a primary care context. From this follow-up research, the researchers found they could visualize the following aspects of the students’ learning process, namely clinical reasoning and reflections in the context of the learning cycle. The research demonstrated that VP models could complement clinical and theoretical teachings and help fill the gaps of traditional education. As a result, students found the new VP model interactive and straightforward, and it also encouraged self-directed learning plus reflective abilities [23].

### 3.4. Consumer Perspectives on Virtual Care

Given the previous sections discussed applications of virtual care and some of the positive aspects of adopting virtual care in varying aspects of healthcare, the following section digs deeper into consumer perspectives, including opinions of both patients and providers, as satisfaction has become an increasingly important measure in the deployment of any solution.

#### 3.4.1. Patient Perspectives

In another study [24], researchers compared “the satisfaction of obstetric patients who received one-third of their antenatal visits in videoconference (virtual care) compared to those who received 12–14 face to face visits in clinic with their physician/midwife.” Satisfaction surveys were sent out to 378 patients from the virtual care cohort and 795 patients who received traditional obstetric services between 2013 and 2015 [24]. The virtual care cohort reported significantly higher overall satisfaction; however, the researchers noted that the virtual care model was selected more by those who were not experiencing pregnancy for the first time [24]. Therefore, one can conclude that although virtual care can provide a great alternative to traditional care, it may not be the most optimal choice for all patients.

Nalverro’s research group [25] implemented a Virtual Health Room (VHR) initiative in a village in Sweden. The study evaluated patient perceptions and the usability of the VHR as well as its contributions to their healthcare. Study participants were asked to complete a 13-question version of the Patient Activation Measure (PAM), patients’ demographic information, as well as a questionnaire that asked for information about their satisfaction with their VHR visit [25]. The researchers found that “respondents with lower PAM scores were less satisfied with the technical performance of the VHR, but equally likely to think the VHR made a good contribution to access to healthcare...In contrast, older patients were less likely to value the contribution of VHR, but no less likely to be satisfied with its technical performance...There were no relationships between level of education and distance travelled and perceptions of the VHR” [25]. Again, even though virtual care services have overall been well accepted by patients, they can never address the needs of all individuals.

A final example to provide patient perspectives of virtual care is in the area of postoperative education, discharge instructions, and follow-up appointments for new ostomates post-discharge. White et al. [26] studied 10 patients who attended two outpatient virtual visits following hospital discharge. The virtual visits were completed through advanced video conferencing software. “All patients successfully completed two virtual visits from their homes... 90% felt these visits helped with ostomy management and agreed they should be part of a discharge plan...All patients felt comfortable with a virtual format...Common themes discussed included poaching and skin irritation” [26]. The researchers concluded that virtual visits are feasible, and patients are also satisfied with the virtual format when addressing certain types of care goals. As mentioned before, virtual visits may be very beneficial for consult-type visits, but they are still limited if physical assessments are required [26].

#### 3.4.2. Provider Perspectives

A group of investigators at the Nemours Children’s Health System launched a Nemours App for Asthma as part of their new digital strategy to provide tools in a single platform. “This kind of an app provides immediate access, another thing that millennial consumers look for. And then we also measure costs and people and growth. We are looking at one or two other chronic conditions that we would want to market, but immediately thereafter we are going to work on a well child. This is how families have a healthcare coach in the palm of their hand” [27]. The app helps clinicians to create an asthma plan for asthma patients in their patient record; however, the plan will only be provided to the family unless clinicians put the plan into the notes section for families to easily access and print out at home [27]. At first, clinicians anticipated themselves to be overwhelmed with extra work they needed to complete; however, the results of the research suggest that not only was the volume manageable, but it was also an effective way to address questions from patients and their families [27]. “It became apparent how absolutely critical telehealth was going to be to provide instant access to a physician who could see and hear the child, and provide guidance in the case of early exacerbations” [27].

In another study, Mammen et al. [28] discussed physician perceptions of virtual visits for Parkinson’s disease (PD). Delivering care remotely has become increasingly valuable and beneficial; however, there is a lack of qualitative data from both patients and providers on how they perceive virtual visits; therefore, the researchers conducted a qualitative analysis about how patients and providers perceived virtual visits related to Parkinson’s disease. “The sentiment analysis for patients was strongly favorable (+2.5) and moderately favorable for physicians (+0.8). Physician scores were lowest (−0.3) for the ability to perform a detailed motor examination remotely” [28]. As a result, although both patients and providers perceived virtual visits favorably, technical challenges varied on a case-by-case basis, which affected their satisfaction with remote care [28].

The results of this scoping review suggest there is an overall positive reaction to virtual care approaches by healthcare providers and patients. Based on this review, it can be stated that virtual care, whether in the form of video conferencing technology that supports remote consultations or as a supplementary care tool in the form of virtual reality, has expanded to many differing aspects of healthcare. This expansion has influenced pain and anxiety management, virtual consultations and follow-up visits, rehabilitation and therapy services, outpatient clinics, and emergency services. Although the findings in each of these areas have not proven virtual care to be a significantly effective intervention, the recipients of care have mostly expressed their interest to incorporate virtual care services into their future care. Technological barriers cannot be neglected either, as with the elderly and frail population, virtual care may not be the most suitable; hence, millennials will be the drivers for the future of virtual care.

## 4. Discussion

From the scoping review, it can be seen that virtual care refers to the use of video technology to connect patients and healthcare providers together regardless of where each party is located. The use of this technology has also influenced a variety of clinical settings, such as pain management, mental health, rehabilitation and therapy services, outpatient clinics, and emergency services. With mostly positive perceptions of the technology, virtual reality technology has also become a part of virtual care by augmenting the virtual care experience. “VR technology has become increasingly affordable, flexible and portable, enabling its use for therapeutic purposes in both inpatient and outpatient environments” [20]. The expansion of tools and technology that support virtual care have complemented the traditional “bricks and mortar” medical practice while at the same time improving healthcare education. Remote training of health professionals using online modules or simulations of virtual patients have been well received by students, with the majority of health professionals expressing interest in using a similar mode of training in the future or applying virtual care to their practice. As Shumaher et al. [29] state, “It is known that the creation of a web-based educational system consists not only in the digitization of texts or printed materials, but in providing its own language, principles, tools and methods, which makes the virtual learning environment (VLE) a space dynamic and interactive, current, closer to the reality of the user and extremely rich, as it allows the use of different media resources that make the teaching-learning process more creative, interesting and powerful.” Lastly, consumer perspectives on virtual care from both patients and providers have been positive as well, especially among patients. It is undeniable that convenience and efficiency play a great factor in patients’ choice for virtual care. However, the technology can still be a barrier to receiving care for some groups such as the elderly.

To list a few of the efficiencies of virtual care compared to traditional visits, wait time is often decreased, and the need to travel to an onsite location for care is removed, hence also decreasing or eliminating transportation costs. As previously mentioned in the study by Gordon et al. [6], the costs associated with patients visiting in-person health clinics, urgent care centers, emergency departments, and primary care physician visits were estimated to be $36, $153, $1735, and $162 higher in non-virtual visits than virtual visits, respectively. Aside from cost and wait times, the satisfaction level in patients who have received care virtually did not decrease compared to traditional visits. In fact, depending on the type of visit required, patients often expressed interest to include virtual visits in their future care plan. Other efficiencies and benefits that do not directly relate to care can be the environmental impact.

As many are familiar with the 2020 global pandemic caused by coronavirus, this has led to some dramatic changes in the way people live and work [30,31]. Governments all around the world have mandated the closure of non-essential services, such as restaurants and entertainment venues, while essential services, such as clinical programs, pharmacies, public transportation, and grocery stores continue to remain open during the pandemic. Given the risks of the pandemic, many countries also began the practice of social distancing, hence reducing the need to visit a care facility, and having virtual visits where possible became the public’s preference [30,31]. With an increase in virtual visits, the need to travel decreased; therefore, this is a perk in terms of environmental impacts and decreasing the carbon footprint on this planet [32]. Progressively, virtual care can become the norm for many patients and providers, and it may be hard for some to revert back to traditional practice. If it was not for COVID-19, virtual care uptake and expansion may not have occurred at such speeds.

In terms of the implications of virtual care on health informatics practice, we can expect there to be many collaborations between health informatics professionals and healthcare organizations in joint research projects on the implementation of virtual care [30,31]. This can include being a consultant for healthcare organizations as they implement a new virtual care technology or conducting evaluations of the post-implementation of virtual care technologies. If evaluations produce positive outcomes (e.g., [30,31,32,33]), the government may also devote more funding toward future virtual care research or wide-scale implementations. This will also affect the training and education for health informatics programs.

Health informatics is a relatively new program in many universities, and some do not even offer this program. Most students benefit from this program if they have career interests to work in the healthcare sector as an informatics professional. It is common for graduates of the program to become a liaison between technology and medical professionals, as they implement technological solutions required by clinical workflows [34]. Given that virtual care will potentially transform the way care has been delivered, it would be worthwhile to incorporate courses as part of the curriculum that cover the successes and failures of virtual care implementations. As an example, the Health Information Science program at the University of Victoria is planning to add an elective course on virtual care to trial the popularity of this topic prior to further embedding this as part of their mandatory course structure. Contents of the course can entail the implementation process of a virtual care tool, with real-life examples or guest speakers who have the knowledge and experience of virtual care implementations [31]. Future learning experiences in work settings terms can also include opportunities to work on virtual care projects [31], which can ultimately open up new employment opportunities for students.

Limitations of the study include the selection of articles. The majority of the articles selected were from the United States or other countries around the globe, with very little from Canada. Therefore, the results of the scoping review reflect the status of virtual care elsewhere and will not accurately reflect its state in Canada. Given this is a scoping review, often the more articles included, the less biased the results will be. Even though the study began with a search of 111 articles, only 28 articles were included in the end results. Hence, the accuracy of this study can be improved if more articles were included.

Future research can expand on the knowledge built from the scoping review to conduct an observational research study on the implementation of virtual care technologies at a clinic, hospital, or even at a broader level, such as healthcare organizations in Canada. This will allow oversight from initiation of implementation to the final uptake of the virtual care technology by end users. Both challenges and benefits of the implementation process can be documented, along with the drivers for implementation, and the goals to be achieved as a result. Upon uptake, end user satisfaction can also be measured to produce a comprehensive research of a virtual care implementation project, which can ultimately provide a foundation for additional virtual care projects in Canada given one of the limitations of this study was the lack of Canadian articles.

## 5. Conclusions

By connecting patients and providers remotely, virtual care has made its presence in different aspects of healthcare, including healthcare education, as well as positive impressions from end consumers. As technology advances, virtual care becomes more accessible, and utilization will increase; hence, acceptance of virtual care and the technology that supports it can also increase in parallel. The technology supporting virtual care, such as video conferencing tools and virtual reality, will also mature to enhance patient and provider experience in using those tools.

## Figures and Tables

**Table 1 healthcare-09-01325-t001:** Keywords for database search.

Virtual Care	Online Care	Mobile Doctor	Devices	Telehealth
Video Conference	Monitoring	Virtual Clinics	WebRTC	Distance
Telemedicine	Technology	Digital Health	Virtual Visits	Application

## Data Availability

Not applicable.

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
