# Peer review of "Connecting the World of Healthcare Virtually: A Scoping Review on Virtual Care Delivery"

_healthcare, 2021, doi:10.3390/healthcare9101325_

Round 1

Reviewer 1 Report

This paper provides valuable information that contributes to knowledge, the authors have appropriately prepared this paper. However, authors should address all comments/concerns to ensure that all problems affecting the sobriety of the research are fixed.

  • Abstract: It should be written more clearly. The authors should explain the purpose, methods and findings (even if not numerical outcomes) in a simple, comprehensive and smooth manner. The authors indicated purpose and methods but require improvement.
  • Materials and Methods Section: We think this section needs more detail, for example, data checking, assess the eligibility of the data … etc.
  • This paper contains a lot of sentences directly quoted from other research legitimately, however, we suggest reducing some of these sentences during paraphrasing to make the research more distinct and coherent.
  • Figures: Figure 1 is unclear and blurry.
  • English Writing: This paper requires moderate proofreading of the entirety of the paper to eliminate all the issues associated with typos, spelling, and grammar mistakes. There are a lot of writing issues so we recommend the authors to check the paper closely because without clear English writing the paper will be incomprehensible or unreadable.
  • List of references: All references are related to the paper topic as well as recent. However, it should follow accurately the MDPI-Healthcare style. Some search names in the references list begin with an uppercase letter in each word such as [5], [6] … etc. and other words begin with a lowercase letter, such as [1], [5] … etc. Authors should standardize the writing style of research names. Journal names should be in italics such as [7], [28]… etc. This paper requires a moderate check of the reference list.

Author Response

Thank you for your review! Below are changes that have been addressed.

Abstract: as suggested by the reviewer, I made the abstract more coherent and succinct, by including the purpose and questions this scoping review strives to answer before describing the review methodology and framework used.

Materials and Methods section: due the fact that a specific framework was followed for the scoping review, assessing eligibility of the data was not part of the procedures. I do agree that this is a very valid point, however because this is a scoping review with literature taken from accredited sources, these details were not included.  Figure 1 has been replaced with a higher resolution image.

English Writing: another round of proof reading has been completed to fix all grammar errors and typos.

List of References: edited to ensure the styles are coherent throughout the paper.

Reviewer 2 Report

The title and aims of this article are somewhat misleading. It should be explicitly mentioned that this is a scoping review. Please see below for my other comments.

Specific comments:

  1. The abstract should be a total of about 200 words maximum. The abstract should be a single paragraph and should follow the style of structured abstracts, but without headings. Please also remove references (unless absolutely essential) from the abstract.
  2. "People often think healthcare as being carried out in “brick and mortar” institutions that provides remedies for patients in a specific building or location" - this statement is acceptable for an editorial or a perspective piece but not for this type of article (e.g., original research or reviews). It should also be "provide" and not "provides".
  3. "keyword 1; virtual care 2; virtual clinics 3; healthcare 4; virtual reality" - please recheck the keywords. Please provide three to ten relevant keywords.
  4. Please clearly specify the inclusion and exclusion criteria for the review.
  5. Was the review protocol prospectively registered as per PRISMA-ScR guidelines?
  6. Please change "does not involved" to "does not involve".
  7. It is unclear how many investigators were involved in the study selection and abstraction process and how were disputes regarding the inclusion/exclusion of studies resolved? The methods used were not adequately described; exactly who did what to identify, review, assess and resolve disagreements in the identified manuscripts. More details are required. 
  8. In Figure 1, please use the latest PRISMA flowchart template and do provide the reasons for exclusion in the diagram.
  9. The legal framework for virtual care is very pertinent and should be discussed. Despite all the purported benefits, we know that the utilization rates of remote services remain low globally consequent upon legal uncertainty, payment issues and technical challenges, as well as patient and physician unfamiliarity (citation: pubmed.ncbi.nlm.nih.gov/32635633).
  10. A wide range of clinical scenarios found teleconsultations to be clinically useful but potentially limited to more straightforward clinical interactions, for example, teleconsult may not be appropriate for breaking bad news (citation: ncbi.nlm.nih.gov/pmc/articles/PMC7288637).
  11. "In addition, as Wilson [27] describes, while the older population may receive the majority of a country’s healthcare dollars, the future of virtual care will undoubtedly lie with the younger generations to drive the trends" - the conclusion should summarise the key points of the article rather than introduce new discussion points. This should be restructured.
  12. Funding information was incomplete.

Author Response

Thank you for your review! Below are changes that have been addressed.

  1. The abstract has been reduced to 198 words, and the reference has also been removed.
  2. This sentence was removed from the abstract. Other typos and grammar errors were also reviewed and updated in the abstract.
  3. Did not find that the keywords required any updating, unless I am misunderstanding the ask here.
  4. Articles were included if they touched on any one of the research questions and excluded otherwise. Articles were also excluded if they were not studies.
  5. Unsure what is meant by prospectively registered, but the majority of the requirements in the PRISMA-ScR checklist were satisfied, with the exception of "protocol and registration"
  6. Update made
  7. Two investigators were involved, where we each reviewed articles separately on Covidence. For articles that had differing opinions, a meeting was setup to finalize the decision.
  8. Updated the PRISMA diagram with the 2020 PRISMA flow diagram: http://prisma-statement.org/prismastatement/flowdiagram.aspx
  9. The legalities of virtual care are discussed in the introduction, where it is important to consider data residency, especially when the technology captures PII or PHI.
  10. Agreed. Virtual care definitely is not a "catch-all" solution, but does provide some convenience for certain clinical scenarios. For example, I've recently reach in contact with a clinic that has highly immune compromised patients. They reached out requesting virtual care to do remote assessments of their patient's skin for injections, as its not suitable for them to come into the hospital.
  11. This was removed, as it was also noted by another reviewer.
  12. Funding information has now been populated. No funding were applicable for this scoping review.

Reviewer 3 Report

Dear authors,
Thank you for letting me read your manuscript `Connecting the World of Healthcare Virtually ´ which I found interesting and valuable for health care workers in many fields.
I will recommend you to revise your text to make it more readerfriendly. A short list of suggestions will follow below:
Abstract
I suggest you follow the common rules for an abstract and clarify knowledge gaps, aims and methods (i.e. I did not understand it was a literature study) and clear answers to your research questions. And remove references in this text,
Introduction
The last section starting with ´Hence, the objective (note, there are several).........of this research (actually a paper in my eyes), is loose and not connected to a knowledge gap and the reader don´t understand why you are doing this work.
Methods
What kind of research did you plan to do? Please, tell the reader before you mention the methods you choose. 
Results
In part 3.2.4, please consider removing the many abbrevations and unnecessary detailed numbers e.g, percentages of patients. And I lack the age of the focus population in the research studies.
Discussion
In summary and what I guess is (or should be) your discussion of your findings/results there are references given so I am not sure if it is your conclusion or the authors of the reference paper. Please, clarify.
Conclusions
The same applies as in the abstract, there is little to connect to your objectives. Also, the text contains discussion and references. That is uncommon in scientific writing. Please, consider a major revision of this part.

Author Response

Thank you for your review! Below are changes that have been addressed.

Abstract: this was cleared up to clearly include the objectives of the paper, and they type of research done (scoping review) to address the research questions. References were also removed from this section.

Methods: Scoping review was the type of research done, as mentioned in the first sentence of this section. This was also included in the abstract, which should now make this section more clear.

Results: in 3.2.4, all the unnecessary acronyms were removed, and the percentages were removed as well. The only percentages that were kept were in the results summary of this section. I dont believe the age of the population was factored into their study.

Discussion: the discussion was written with references as results from the references back up my own thoughts. Apologies for the confusion.

Conclusion: after changes have been made to the abstract, the conclusion should make more sense now, as it summarizes the answers to the research questions introduced earlier. The reference was removed from this section.

Round 2

Reviewer 2 Report

In the title, it should be explicitly mentioned that this is a scoping review on virtual healthcare delivery.

Author Response

The title has been adjusted to clearly state this is a scoping review